# BRAF Gene and Melanoma: Back to the Future

**DOI:** 10.3390/ijms22073474

**Published:** 2021-03-27

**Authors:** Margaret Ottaviano, Emilio Francesco Giunta, Marianna Tortora, Marcello Curvietto, Laura Attademo, Davide Bosso, Cinzia Cardalesi, Mario Rosanova, Pietro De Placido, Erica Pietroluongo, Vittorio Riccio, Brigitta Mucci, Sara Parola, Maria Grazia Vitale, Giovannella Palmieri, Bruno Daniele, Ester Simeone

**Affiliations:** 1Department of Clinical Medicine and Surgery, Università Degli Studi di Napoli “Federico II”, 80131 Naples, Italy; pietrodep91@gmail.com (P.D.P.); erica.pietroluongo@gmail.com (E.P.); vittorioriccio1990@gmail.com (V.R.); brigitta.mucci@gmail.com (B.M.); saraparola3@gmail.com (S.P.); 2Oncology Unit, Ospedale del Mare, 80147 Naples, Italy; laura.attademo@gmail.com (L.A.); davidebosso84@gmail.com (D.B.); cinzia.cardalesi@gmail.com (C.C.); rosanovamario@hotmail.com (M.R.); b.daniele@libero.it (B.D.); 3CRCTR Coordinating Rare Tumors Reference Center of Campania Region, 80131 Naples, Italy; marian.tortora@gmail.com (M.T.); giovpalm@unina.it (G.P.); 4Department of Precision Medicine, Università Degli Studi della Campania Luigi Vanvitelli, 80131 Naples, Italy; emiliofrancescogiunta@gmail.com; 5Unit of Melanoma, Cancer Immunotherapy and Development Therapeutics, Istituto Nazionale Tumori IRCCS Fondazione Pascale, 80131 Naples, Italy; curvietto.ma@gmail.com (M.C.); dott.mariagrazia.vitale@gmail.com (M.G.V.); ester.simeone@gmail.com (E.S.)

**Keywords:** melanoma, BRAF mutation, immunotherapy, targeted therapy

## Abstract

As widely acknowledged, 40–50% of all melanoma patients harbour an activating BRAF mutation (mostly BRAF V600E). The identification of the RAS–RAF–MEK–ERK (MAP kinase) signalling pathway and its targeting has represented a valuable milestone for the advanced and, more recently, for the completely resected stage III and IV melanoma therapy management. However, despite progress in BRAF-mutant melanoma treatment, the two different approaches approved so far for metastatic disease, immunotherapy and BRAF+MEK inhibitors, allow a 5-year survival of no more than 60%, and most patients relapse during treatment due to acquired mechanisms of resistance. Deep insight into BRAF gene biology is fundamental to describe the acquired resistance mechanisms (primary and secondary) and to understand the molecular pathways that are now being investigated in preclinical and clinical studies with the aim of improving outcomes in BRAF-mutant patients.

## 1. Introduction

The identification of signalling pathways and tumour immune microenvironment interactions has revolutionized the treatment of melanoma [1]. To date, two largely mutually exclusive groups of cutaneous melanomas can be categorised: those harbouring an activating BRAF mutation (mostly BRAF V600E), which represent 40–50% of all melanoma patients, and those harbouring other mutations than BRAF [2]. Indeed, as widely acknowledged, the identification of the RAS–RAF–MEK–ERK (MAP kinase) signalling pathway and its targeting has represented a valuable milestone for the advanced and, more recently, for the completely resected stage III and IV melanoma therapy approaches [3]. In patients with unresectable or metastatic disease, the therapeutic available strategies of targeted therapy (TT) with BRAF inhibitor (BRAFi) and MEK inhibitor (MEKi) and immune check-point inhibitors (ICIs) have led to a significant improvement in overall survival (OS) and progression-free survival (PFS). However, despite the recognized progress, the two different approaches approved so far for metastatic disease allow a 5-year survival of no more than 60%, even with important differences according to first-line drug(s) used and prognostic factors; moreover, most patients relapse during treatment due to acquired mechanisms of resistance [4,5]. Deep insight into the BRAF gene biology is fundamental to further describe the resistance mechanisms (both primary and secondary) and to understand the molecular pathways that are now being investigated in preclinical and clinical studies with the aim to improve outcomes in BRAF-mutant patients. In this review, starting from past knowledge about BRAF gene biology, we report the most up-to-date information about BRAF mutational assessment and the most up-to-date understanding of clinical trials results, trying to shed light on the therapeutic approaches of the future in BRAF melanoma-mutant patients.

## 2. BRAF Gene Biology

The BRAF gene is located on chromosome 7 (7q34) and encodes the BRAF protein, a 94 kDa intracellular enzyme of 766 amino acids involved in the mitogen activated protein kinase (MAPK) pathway [6]. The MAPK pathway consists of a chain of intracellular proteins which regulates physiological cell functions including growth, differentiation, proliferation and apoptosis [7]. BRAF, as well as ARAF and CRAF, is a MAPK kinase kinase (MAPKKK), and it is typically activated by GTPases proteins—namely, RAS proteins—downstream from cell surface receptors, such as EGFR (Epidermal Growth Factor Receptor) or KIT, even if several more kinds of stimuli could lead to its activation [8].

BRAF protein consists of three conserved regions (CRs): CR1, which is composed of a RAS-binding domain (RBD) followed by a cysteine-rich domain; CR2 is a serine/threonine-rich domain containing a 14-3-3 binding site; and CR3 is the catalytic serine/threonine protein kinase domain [9]. To be effective, BRAF must dimerize in order to form complexes with MEK, adjuvated by 14-3-3 proteins; the latter ones are involved in both active and inactive phases of BRAF signalling [10].

Once activated, BRAF phosphorylates mitogen-activated protein kinase/extracellular signal-regulated kinase ERK kinase (MEK) which, in turn, phosphorylates the extracellular signal-related kinases 1 and 2 (ERK1/2) [11]. ERK proteins are the last effectors of the pathway: once phosphorylated, they dimerize and translocate into cell nucleus, thus activating, through phosphorylation, many transcription factors, such as c-Jun and c-Myc [12]. The final goals of this signalling pathway in physiological conditions are the control of cell cycle progression and the regulation of apoptosis [11] (Figure 1).

## 3. BRAF Mutations in Melanoma: Epidemiology and Clinic-Pathological Correlations

Since the early 2000s, BRAF has been identified as a commonly mutated gene in human tumours [2]. Mutations in the BRAF gene could cause an impaired protein function, depending on localization and type [13].

Concerning cutaneous melanoma, the most frequent (65%) and relevant alterations in BRAF gene sequence are those affecting codon V600 (formerly named V599) in the exon 15 [2,14]. BRAF V600 mutations have been detected in nearly 50% of all cutaneous melanoma patients [15]. Non-V600 mutations, which are less frequent than V600 ones, have been found in 11% of all cutaneous melanoma patients [14].

BRAF mutations in cutaneous melanoma are most common on the trunk (affecting less frequently the head and neck), on skin without marked solar elastosis and in younger age, thus suggesting a physiopathology role for intermittent UV exposition in early life rather than chronic sun damage [16]. A recent study using sequencing data showed a model of the propagation and selection of clones with different categories of BRAF mutations to establish their evolutionary trajectories. The phylogenetic trees of cutaneous melanoma samples with amplification of BRAF express a major dominant clone, with only rare intermediates that are persistent from the previous selective sweeps, consistent with a linear evolutionary process. However, it is still not clear whether melanoma with amplification of BRAF experiences iterative selective sweeps and, if so, what the underlying molecular basis of this process might be [17].

Actually, clinicopathological characteristics and frequency are different for each kind of BRAF mutation, so we should consider them as different entities.

### 3.1. BRAF V600 Mutations

BRAF V600E is, globally, the most frequent mutation observed in cutaneous melanoma patients, accounting for 70–88% of all known V600 BRAF mutations [16]. It consists of an amino acid change from valine (V) to glutamic acid (E), resulting in a 480-fold increase in kinase activity (catalytically active conformation) compared with native protein [13]. BRAF V600K is the second most common mutation (10–20% of all V600 BRAF mutations) in cutaneous melanoma and, as V600E mutation, it consists of an amino acid change, with a valine (V) replaced by a lysine (K) [18]. Other rarer codon V600 BRAF mutations, approximately 10% of all V600 mutations, are V600R (<5%), V600D (<5%), V600E2 (<1%), V600M (<1%) and V600G (<1%) [17,19,20]. Cutaneous melanomas harbouring BRAF V600E and V600K mutations, even if similar from a molecular point of view, have distinct clinicopathological features (Table 1). In fact, BRAF V600K-mutant cutaneous melanomas are considered more aggressive than V600E ones, since they have shown less tumour regression and shorter progression-free survival during treatment with combined BRAF and MEK inhibitors, together with a shorter disease-free interval from diagnosis of primary melanoma to the occurrence of first distant metastasis [21,22]. Analysis of BRAF V600K-mutant cutaneous melanoma samples from the Cancer Genome Atlas highlighted, with respect to V600E, an upregulation of energy metabolism, emphasizing their clinical aggressiveness [23]. On the other hand, an older age at diagnosis, a higher degree of cumulative sun-induced damage and a higher mutational burden have been described in V600K-mutant cutaneous melanoma with respect to V600E melanomas, explaining good response to immunotherapy [21,22].

Concerning prognosis, BRAF V600 mutations are statistically significantly associated with reduced OS in cutaneous melanoma patients [24]. In fact, it must be stressed that there is a higher trend of BRAF V600-mutant melanoma, with respect to BRAF wild-type ones, to involve the brain and liver as a first site of metastasis, thus affecting negatively the prognosis of these patients [25]. BRAF V600 mutations also have an important predictive role, since metastatic cutaneous melanoma patients harbouring them do respond to specific inhibitors, as further described. Rare V600 BRAF mutations, such as V600R, V600D and V600M, have been associated with good response to BRAF inhibitors and acceptable OS compared to V600E/K-mutant melanoma patients [26].

### 3.2. BRAF Non-V600 Mutations

BRAF non-V600 mutations, as stated before, are less frequent than V600 ones, and their prognostic and predictive role is, to date, still difficult to elucidate.

L597, K601 and G469 mutations, also known as class II BRAF mutations, determine an increased kinase catalytic activity, different from monomeric V600-mutant protein, through constitutive dimerization. Interestingly, they are located in different regions of the gene: L597 and K601 in the activation segment, whilst G469 in the glycine rich region of BRAF [14]. Even if these mutations do not confer sensitivity to BRAF inhibitors, they activate downstream target proteins, thus explaining sensitivity to MEK inhibitors [27].

Codon D594 and G596 mutations, also known as class III BRAF mutations, have been described as kinase-impairing alterations [28,29]. In fact, different from V600E mutations, which cause hyperactivation of downstream kinase pathways, kinase-impairing mutations cause a reduction in BRAF catalytic activity; these proteins are RAS dependent and have low or absent kinase activity [14,30]. Codon D594 and G596 mutations are rare, < 4% of all melanoma patients, but have been associated with a good prognosis and a more prolonged OS than V600-mutant patients [30]. Codon N581 mutations, very rare, are also kinase-impairing mutations [29].

### 3.3. BRAF Fusions

Oncogenic BRAF fusions are the result of genomic rearrangements which constitutively cause activation of BRAF kinase catalytic activity through the loss of the auto-inhibitory domain of the gene, being replaced by another gene in 5′ position [31]. BRAF fusions are estimated to occur in 3–6% of all melanoma patients, with a higher frequency in female gender, younger age and certain histopathologic subtypes such as spitzoid melanomas [32,33]. The location of the breakpoints occurring in introns 7–10, thus preserving the kinase domain, and more than 40 partner genes have been identified, most of them being on the same chromosome of the BRAF gene; the moderate UV signature observed in tumour samples harbouring BRAF fusions suggests that they are not a consequence of UV exposure [32].

## 4. BRAF Mutational Assessment: State of Art of Companion Diagnostic and Laboratory-Developed Tests

The BRAF mutational assessment landscape in melanoma patients has evolved over time, and the modern scenario is characterized by the availability of several companion diagnostic and laboratory-developed tests [34]. Moreover, in most cases, a not negligible advantage is represented by abundant biopsy material for molecular tests, coming from a variety of primitive tumour excision or metastatic biopsies [35]. Recently, the easily reproducible tool of liquid biopsy has also found applicability in melanoma patients, especially for monitoring therapeutic response [36,37,38]. Indeed, it has been shown that in patients with advanced BRAF mutated melanoma undergoing treatment with TT, higher levels of plasma circulating tumour DNA (ctDNA) may predict disease progression earlier than imaging and/or clinical assessments [39].

### 4.1. Immunohistochemistry

Immunohistochemistry (IHC) is a consistent option for detecting the BRAF exon 15 p.V600E mutation [40], since it is simple and low cost with rapid turnaround time (TAT) and high sensitivity and specificity. IHC can also be applied to tissues with low tumour content that are not suitable for DNA-based molecular analyses; it produces clear visualization of the condition of the entire tumour [41]. However, its main limitations are represented by the possibility of false-negative results due to heterogeneity or a low concentration of *BRAF* exon 15 p.V600E and the inability to identify *BRAF* exon 15 other variants, such as the V600 K one [41]. The most utilized antibody is the monoclonal antibody VE1 [42], which showed high sensitivity and specificity (97% and 98%, respectively) when compared with a DNA-based approach [43], with pyrosequencing (85% and 100%, respectively) and PCR-based approaches (98.6% and 97.7%) [44,45]. Nevertheless, the antibody VE1 has a low limit of detection and allows for the detection of BRAF p.V600E mutated cells at the single-cell level. Moreover, since the interpretation of staining results can be complicated in tissue slides rich in melanin pigmentation, the use of a red-coloured immunostaining or Giemsa counterstaining should be taken into account [46]. Additionally, a recent study investigating the reproducibility of IHC using the VE1 antibody in a series of acral melanoma obtained results that were comparable with standard molecular analyses. Interestingly, the authors found remarkable clear proliferations of sub-clones with different BRAF status in the same sample of acral melanoma (intra-tumour heterogeneity), as well as detected different BRAF status comparing some metastatic lesions of acral melanoma with primary lesions (inter-tumour heterogeneity), highlighting that the exclusive inclusion of acral melanoma might have contributed to the high rate of heterogeneity reported [47].

### 4.2. Sanger Sequencing

Sanger sequencing represents a valid option for melanoma patients, since, for point mutations and small variant detection, it is an easily available, reproducible and relatively low-cost approach. Its main limitation is the low sensitivity [48]. In melanoma patients, when compared with the real-time PCR (RT-PCR) approach cobas^®^ 4800 BRAF V600 mutation test (Roche, Basel, Switzerland), the Sanger sequencing achieved an overall agreement ranging from 95.2% to 97.7% [49,50]. Moreover, Sanger sequencing should be used in cases of a negative cobas^®^ 4800 BRAF V600 mutation test result, since it performed better than the latter test (43% and 35%, respectively) [51].

### 4.3. Pyrosequencing

Conversely to IHC, which is unable to identify BRAF exon 15 non-V600 other mutations, pyrosequencing finds its applicability in precisely detecting BRAF non-V600E mutations. Indeed, a higher rare BRAF mutations detection rate has been reported for pyrosequencing (92.9%) than the cobas^®^ 4800 BRAF V600 mutation test (50.0%) and IHC (21.4%) [19], highlighting the utility of pyrosequencing in detecting rare BRAF mutations, which otherwise should be excluded from TT approaches.

### 4.4. Real-Time PCR

For detecting BRAF mutations in melanoma patients, the real-time PCR (RT-PCR) approach utilizes a set of primers: one for targeting BRAF mutations and another for identifying the wild-type sequence [52]. The previously mentioned cobas^®^ 4800 BRAF V600 mutation test or THxID-BRAF kit is the Food and Drug Administration (FDA) approved RT-PCR test for detecting BRAF exon 15 p.V600 in melanoma patients [53]. When compared with Sanger sequencing, a 100% of success rate was reported for cobas^®^ 4800 BRAF V600 mutation, whereas a failure rate of 9.2% was reported for Sanger sequencing [54]; when compared with several other techniques, such as Sanger sequencing, pyrosequencing and allele-specific PCR, the frequency of BRAF exon 15 p.V600 mutations was marginally higher for the other techniques than for the cobas^®^ 4800 BRAF V600 mutation test (35.7% vs. 34.0%, respectively). However, wild-type cases were assessed with a higher frequency by the cobas^®^ 4800 BRAF V600 mutation test (63.3% vs. 62.9%) [55], highlighting the relevant clinical efficacy of the cobas^®^ 4800 BRAF V600 mutation test.

### 4.5. Next-Generation Sequencing

Next-generation sequencing (NGS) is a technology with higher sensitivity but also with higher costs and longer TAT compared to allele-specific tests, whose application in melanoma patients should be limited to those cases displaying a negative result with allele-specific BRAF exon 15 V600E/K PCR [56]. Since, as previous stated, the main limitations of NGS are the costs and TAT, these should be bypassed by including the analysis of other genes. Indeed, as reported by Unamuno Bustos et al., about 85% of the entire cohort of the investigated melanomas presented at least one mutation [57], with 50% of cases harbouring a BRAF mutation. Moreover, NGS may also be valuable for identifying rare actionable mutations that are not usually detected by targeted methods, as recently reported in literature, where NGS was able to assess a rare variant of BRAF exon 15 V600E (c.1799_1800TG > AA) ignored by a RT-PCR approach [58].

## 5. BRAF Mutations as Pharmacological Targets in Melanoma

To date, patients with BRAF-mutant melanoma can be treated with TTs and ICIs. The FDA and the European Medicines Agency (EMA) approved three BRAF and MEK inhibitor combination TTs for patients with unresectable/metastatic BRAF mutated melanoma: dabrafenib plus trametinib, vemurafenib plus cobimetinib and encorafenib plus binimetinib. Three ICIs—ipilimumab, nivolumab, pembrolizumab—are also approved alone or in combination in this setting of patients. Results of the pivotal trials for the adjuvant setting are briefly illustrated below.

### 5.1. Dabrafenib Plus Trametinib (COMBI-d and COMBI-v)

The combination of dabrafenib (D) and trametinib (T) was investigated in two prospective, randomized, controlled, phase 3 trials (COMBI-d [59] and COMBI-v [60]). The recent 5-year pooled analysis including 563 treatment- naïve patients with BRAF V600E/K mutated, unresectable or metastatic melanoma were randomly assigned to receive either D + T or D plus placebo or vemurafenib (V). In the pooled patient cohort, the median PFS for D + T was 11.1 months (95% confidence interval (CI) 9.5–12.8) in the intention-to-treat population. The PFS rate at 5 years was 19%. The 5-year PFS rate was higher for patients with normal levels of lactate dehydrogenase (LDH), whilst the 5-year PFS rate was 8% for patients with elevated LDH levels. Furthermore, with a median OS of 25.9 months (95% CI 22.6–31.5) and a 5-year OS rate of 34% (95% CI 30–38), patients with normal LDH levels showed a 5-year OS rate of 43% compared with only 16% for those with elevated LDH levels at baseline. Higher 5-year rate for both OS and PFS was registered in patients with normal LDH basal levels and less than three metastatic disease sites. Interestingly, patients who achieved complete response (CR) reached a 5-year PFS rate of 49% (95% CI 39–58) and OS rate of 71% (95% CI 62–79) [61]. The median duration of CRs was 36.7 months (95% CI 24.1–not reached), highlighting the possibility of resistance also in patients showing deep responses.

### 5.2. Vemurafenib Plus Cobimetinib (coBRIM)

Vemurafenib was the first BRAF inhibitor to be approved by the FDA for the treatment of advanced BRAF exon 15 V600E-mutant melanoma patients [62]. The coBRIM study was the first prospective, randomized, double-blind, Phase 3 trial comparing the combination of vemurafenib (V) plus cobimetinib (C) with V plus placebo [63]. The most updated results showed a median PFS for the combination arm of 12.3 months (95% CI 9.5–13.4) and a median OS of 22.3 months (95% CI 20.3–not estimable). The 2-year OS rate was 48.3% (95% CI 41.4–55.2). As previously reported for D + T, patients with normal baseline LDH levels had a higher median PFS of 13.4 months (95% CI 12.0–16.5). Regarding the overall response rates (ORR), an objective response was obtained in 70% of the patients, with a CR in 16% of patients and a median duration of response (DOR) of 18.1 months for patients who achieved a CR [64]. A pooled analysis of four randomized clinical trials (BRIM-2; BRIM-3; BRIM-7; coBRIM) revealed LDH levels and the sum of the longest diameters of target lesions as significant baseline characteristics associated with survival outcome in patients receiving the combination TT [65].

### 5.3. Encorafenib Plus Binimetinib (COLUMBUS)

The efficacy of the combination TT of the BRAFi encorafenib (E) and the MEKi binimetinib (B) was investigated in the COLUMBUS trial, a two-part, randomized, open-label, Phase 3 study [66,67]. Encorafenib has interesting pharmacokinetic features, with a 10-fold longer half-life of dissociation (>30 h) than either D or V, enabling persistent target inhibition. The combined therapy with E and B was compared with V monotherapy in the first part of the trial, achieving, after a median follow-up for OS of 36.8 months, a median OS of 33.6 months, compared with 16.9 months for V (hazard ratio (HR) 0.61, 95% CI 0.47–0.79; *p* < 0.0001). With regard to OS subgroup analyses, the E + B combination showed advantage in most but not all subgroups (not in patients with elevated baseline LDH levels). Advantage was detected also for PFS, with a median PFS of 14.9 months for E + B versus 7.3 months for V (HR 0.51, 95% CI 0.39–0.67; *p* < 0.0001). Confirmed ORR was observed in 64% of the patients in the combination therapy arm, who achieved CR in 11% of cases. Liszkay et al. presented updated efficacy results for the combination therapy arm at the annual 2019 American Society of Clinical Oncology (ASCO) meeting, where a median OS for E + B of 33.6 months (95% CI 24.4–39.2) after a median follow up of 48.6 months was shown, as well as a median PFS of 14.9 months (95% CI 11.0–20.2), compared with 7.3 months (95% CI 5.6–8.2) for V monotherapy. The 4-year landmark PFS and OS for E + B was 25% and 39%, respectively [68].

### 5.4. Pembrolizumab and Nivolumab in Monotherapy

Regardless of BRAF mutational status, the anti-programmed cell death protein 1 (PD-1) monotherapies display significant activity in metastatic melanoma. The KEYNOTE-006 study, evaluating the efficacy of pembrolizumab in BRAF V600-mutant melanoma patients, reported a 5-year median OS of 32.7 months (95% CI 24.5–41.6) [69]. Regarding trials evaluating the activity and efficacy of nivolumab, the CheckMate 067 trial (NCT01844505) reported a 5-year landmark OS rate of 44% and a 5-year landmark PFS rate of 29%, whilst in the BRAF-mutant subgroup, a 5-year landmark OS and PFS rate of 46% and 22%, respectively, was shown, with a median OS of 45.5 months (95% CI 26.4–not reached) and a median PFS of 5.6 months (95% CI 2.8–9.5), highlighting a better median OS in this subgroup potentially due to subsequent therapies [70,71]. The CheckMate 037 trial, investigating the efficacy of nivolumab in pre-treated metastatic melanoma patients, reported a median OS of 15.7 months (95% CI 12.9–19.9), with a 2-year OS rate of 38.7% (95% CI 32.8–44.5) in the entire population, with no notable differences in OS in the pre-specified subgroup analyses. No specific OS was reported for BRAF-mutant patients in this trial [71].

### 5.5. Ipilimumab Plus Nivolumab

The CheckMate 067, after a minimal follow-up of 5 years, reported a 5-year OS rate of 52% in all metastatic treatment-naïve melanoma patients treated with the ICIs combination of ipilimumab plus nivolumab [70]. As previous stated, the subgroup analyses showed a slightly better outcome for patients with BRAF mutation, who achieved a 5-year OS rate of 60% compared with 46% of patients without the BRAF mutation (Table 2).

### 5.6. Adjuvant Therapies

Patients with completely resected stage III and IV disease are at high risk of relapse after loco-regional resection, and many could die from metastatic melanoma [72]. Taking into account the efficacy results of combined TT in metastatic melanoma patients with BRAF mutation and the clinical need to improve the outcomes of adjuvant therapy in melanoma, studies have been carried out to establish whether TT in an adjuvant setting would improve outcomes in BRAFV600-mutant patients with resected stage III and IV melanoma. To date, the only combined TT approved by the EMA as adjuvant treatment is dabrafenib plus trametinib, considering the Combi-AD trial results [73]. Indeed, two Phase 3 clinical trials evaluated the efficacy of BRAFi or BRAFi plus MEKi delivered for twelve months in the adjuvant setting. The BRIM-8 trial investigated adjuvant vemurafenib monotherapy versus placebo in patients with completely resected BRAF V600-mutant melanoma, in either American Joint Committee on Cancer (AJCC) 7th edition stage IIC-IIIB (cohort 1) or stage IIIC (cohort 2). Despite a median disease-free survival of 23.1 months versus 15.4 months in vemurafenib and placebo, respectively (HR:0.80, *p* = 0.026), no significant improvement in recurrence-free survival (RFS) was detected in stage IIIC patients, and the study failed to achieve its primary endpoint [74]. The COMBI-AD Phase 3 trial, evaluating the combination of dabrafenib and trametinib in patients with stage AJCC 7th edition IIIA (limited to lymph-node metastasis of >1 mm), IIIB or IIIC cutaneous melanoma resulted in 53% lower risk of relapse than placebo, achieving significant improvement in the 5-year RFS rate (52% versus 36%). Furthermore, the risk of death was 49% lower (HR: 0.51; 95% CI 0.42–0.61), and the risk of distant metastasis or death was reduced by 47%. The percentage of patients who were alive without distant metastasis was 65% for the target therapy arm vs. 54% for the placebo arm (HR 0.55; 95% CI 0.44–0.70) [73,75,76]. Interestingly, the subgroup analysis of RFS showed therapeutic benefits regardless of baseline factors, including AJCC 8th edition disease stage, nodal metastatic burden or tumour ulceration status [72]. The estimated cure rate at a median follow-up of 3.5 years was 54% (95% CI: 49–59) and 37% (95% CI: 32–42) for dabrafenib plus trametinib and placebo, respectively [75]. The favourable survival outcomes achieved by ICIs in Phase 3 clinical trials in advanced melanoma patients have also led to their development in the adjuvant setting, achieving substantial improvements in RFS. The first immunotherapeutic agent approved as adjuvant therapy for resected melanoma in both the USA and Europe, no longer used, was the interferon alpha, characterized by marginal benefit in OS and high frequency of adverse events [77,78]. In 2015, ipilimumab was the first ICI to receive approval by the FDA, but not by the EMA, as adjuvant therapy in patients with fully resected stage III melanoma at high risk of recurrence, based on the results of a randomised, placebo-controlled, Phase 3 trial (EORTC 18071), which reported a 28% reduction of the risk of death and a 24% reduction of the risk of distant metastasis or death for ipilimumab, with grade 3–4 adverse events occurring in 54.1% of the ipilimumab group and 26.2% of the placebo group, respectively [79]. No data are currently available concerning the efficacy of adjuvant ipilimumab therapy according to BRAF-mutation status. The anti-PD-1 ICIs nivolumab and pembrolizumab, for the demonstrated significant improvement in RFS in Phase 3 comparative trials, received FDA and EMA approval for the adjuvant treatment of completely resected melanoma patients [80,81]. In the Phase 3 CheckMate-238 trial comparing nivolumab with ipilimumab or placebo as adjuvant therapy, the nivolumab arm showed significantly longer RFS and better safety profile than adjuvant therapy with ipilimumab, regardless of BRAF status [80]. The 1-year RFS rate was 70.5% in the nivolumab group and 60.8% in the ipilimumab group, respectively. Grade 3–4 adverse events were registered in 14.4% of patients in the nivolumab group and 45.9% of patients in the nivolumab and ipilimumab group. The most up-to-date data reported 4-year RFS rates of 51.7% and 41.2% for nivolumab and ipilimumab, respectively [82]. Analogously, pembrolizumab for high-risk stage III melanoma resulted in significantly longer RFS than placebo, with no new toxic effects reported and no differences according to the BRAF status. The 3-year RFS rate was 63.7% for pembrolizumab versus 44.1% for placebo with a HR of 0.56 (95% CI, 0.47 to 0.68) [83] (Table 3). The combination of nivolumab and ipilimumab has also been investigated in the adjuvant setting. The Phase 2 IMMUNED trial, evaluating nivolumab plus ipilimumab or nivolumab monotherapy versus placebo in patients with stage IV melanoma with no evidence of disease after complete resection or radiotherapy, showed a HR for recurrence of 0.23 (97.5% CI 0.12–0.45; *p* < 0.0001) for the nivolumab plus ipilimumab group versus placebo group, and a HR of 0.56 (0.33–0.94; *p* = 0.011) for the nivolumab group versus placebo. The treatment was discontinued due to adverse events in 62% of patients receiving the combination therapy and 13% in those receiving nivolumab [84]. Other clinical trials are currently ongoing (NCT02656706 and NCT03068455).

## 6. From the Past to the Future of BRAF-Mutant Melanoma Treatment

Despite progresses in BRAF-mutant melanoma treatment, the two different approaches approved for metastatic disease, immunotherapy and BRAFi + MEKi, allow a 5-year survival of no more than 60%, even if with important differences according to first-line drug(s) used and prognostic factors [4,85]. Regarding recently approved adjuvant therapies in radical resected III-IV stage BRAF-mutant melanoma, recurrences occur in fewer than half of patients after 3 years from starting treatment. Looking at survival curves from all these Phase 3 trials, only a small percentage of patients do not benefit from treatment in the very first months with both TT and ICIs. Indeed, with the latter approach, a rapid progressive disease during the first year of treatment could affect as many as one-third of patients; however, a good portion of patients, obtaining partial or complete response with immunotherapy, seem to be long-term survivors, different from patients who start with TT. Understanding mechanisms underlying resistance to currently approved therapies and going back to molecular pathways that are now being investigated in preclinical and clinical studies is necessary to improve outcomes in BRAF-mutant patients (Figure 1).

### 6.1. Combining Immunotherapy with BRAF and MEK Inhibitors

It is well acknowledged that BRAF and MEK inhibitors exert an immunomodulatory effect in melanoma patients: in fact, they promote an immune stimulatory microenvironment by enhancing pro-inflammatory cytokines, i.e., interferon-γ (IFN-γ), and reducing immunosuppressive ones [86], but also increasing T cell infiltration and improving their activity [87]. Supported by preclinical data [88], and given the different pattern and duration of responses, the combination of targeted therapy and immunotherapy has been tested in BRAF-mutant melanoma patients as first-line treatment in advanced disease; however, the results to date are disappointing. Indeed, only one Phase 3 trial has demonstrated a statistically significant improvement in PFS by adding an anti-PD-L1 antibody to BRAF and MEK inhibitors (IMspire150 trial, vemurafenib + cobimetinib ± atezolizumab) [89], whilst the addition of an anti-PD-1 to targeted therapy in a Phase 2 trial did not reach the pre-specified boundary of significance for PFS improvement. Moreover, the OS was similar between the two arms (KEYNOTE-022 trial, dabrafenib + trametinib ± pembrolizumab) [90]. As previously reported, the trial with spartalizumab, an anti-PD-1, added to dabrafenib + trametinib (COMBI-i trial), also failed to demonstrate prolongation in PFS [91]. It must be highlighted that contrary to expectations, even with a short follow-up time, these studies have substantially pointed out a non-synergistic effect during the first months of administration. An increase in long-term survival percentages is probable but needs to be proven. Toxicities are also an issue, with higher G3–4 adverse events in patients undergoing “triplet”. In any case, triple combination therapy (BRAF inhibitor + MEK inhibitor + Anti-PD-(L)1) seems not to be a game changer in this clinical scenario. Modifying a strategy, for example, with intermittent TT or with an induction phase with these drugs followed by immunotherapy, is under investigation in ongoing clinical trials: IMPemBra trial, NCT02625337, and SECOMBIT trial, NCT02631447, respectively. In particular, the SECOMBIT trial is a Phase 2 trial in which patients were randomized in three arms: arm A (encorafenib + binimetinib until PD, followed by nivolumab + ipilimumab), arm B (ipilimumab + nivolumab until PD, followed by encorafenib + binimetinib) or arm C (encorafenib + binimetinib for 8 weeks, followed by ipilimumab + nivolumab until PD, followed by encorafenib + binimetinib); preliminary data from this trial have been presented at the 2020 European Society for Medical Oncology (ESMO) congress, showing non-statistically significant difference in 2-year PFS among the three arms [92].

### 6.2. MAPK Pathway

Re-activation of the MAPK pathway has been found to be present in a high percentage of BRAF-mutant melanomas resistant to BRAF and MEK inhibitors. *BRAF* amplification and *NRAS* and *MEK2* activating mutations are the most frequently observed in both preclinical and clinical settings [93,94]. In this scenario, pan-RAF inhibitors could play a role in reversing resistance to BRAF and MEK inhibitors, as suggested by preclinical studies [95]. Nevertheless, more studies are needed to better understand these mechanisms of resistance and find effective therapies [96]. ERK proteins are the last effectors of the MAPK pathway, as stated before [12]. Mutations in the *ERK* gene are rare in melanoma [97], meaning that enhanced activity of this kinase is strictly dependent on upstream signals. Recovery of ERK activity in BRAF inhibitor resistant melanoma cells was demonstrated just a decade ago, suggesting the role of MEK inhibition in combination with BRAF inhibition to prevent ERK phosphorylation [98]. However, a recent preclinical study on BRAF and MEK inhibitors, resistant BRAF-mutant melanoma cells showed elevated levels of phosphorylation in ERK1/2, consistent with reactivation of the MAPK signalling cascade, this reactivation being independent from MEK activity in most cases [99]. Given the above, ERK has quickly become a target for drug development. To date, several ERK inhibitors are under investigation in Phase 1/2 clinical trials in advanced solid tumours: MK-8353, ulixertinib, ravoxertinib, LTT462 and LY3214996 [100]. ERK inhibitors in advanced *BRAF*-mutant melanoma could be administered after BRAF and MEK inhibitors’ failure or together with these drugs to prevent the onset of resistance; however, concerns about additive toxicities would probably limit the clinical applicability of a “triple inhibition”, even if the predicted efficacy of similarly intensive approaches in *BRAF-V600E*-mutant colorectal cancer is high [101]. On the other hand, MEK and ERK inhibitors have demonstrated synergistic activities in RAS-mutant tumours [102] and, according to preliminary pharmacokinetic data, LY3214996 could be a good partner for MEK inhibitors in future clinical trials, in order to provide adequate tolerability [103]. Phosphoinositide-3-kinase (PI3K) inhibitors, which are described in the next paragraph, could also be a potential partner for ERK inhibitors.

### 6.3. PI3K/AKT/mTOR Pathway and PTEN

As well as the MAPK pathway, the phosphoinositide-3-kinase (PI3K)/protein kinase B (PKB, better known as AKT)/mammalian target of rapamycin (mTOR) signalling pathway is involved in cellular growth, proliferation and survival [104]. PI3K, stimulated by membrane receptors, converts, through its catalytic domain, phosphatidylinositol (3,4)-bisphosphate (PIP2) on the plasma membrane into phosphatidylinositol (3,4,5)-trisphosphate (PIP3), which is the docking site for AKT. After its binding, AKT can activate mTOR complex (mTORC), leading to protein synthesis and cellular proliferation by stimulating 4E binding protein 1 (4EBP1) and ribosomal protein S6 kinase 1 (S6K1) [105]. Dysregulation of this pathway is frequent in metastatic melanoma [106]; however, mutations of genes involved in this pathway are rare [107], meaning that other molecular mechanisms can activate PI3K-AKT signalling, i.e., *NRAS*, *c-KIT* and *HER4* activating mutations [106]. Loss of phosphatase and tensin homolog (PTEN), which catalyses dephosphorylation of PIP3 in PIP2 thus inactivating the PI3K pathway, has been detected in no more than 30% of melanoma patients [108]; intriguingly, loss of PTEN is never associated with *NRAS* mutations in melanoma, whilst it is frequent in *BRAF*-mutant ones [109]. PTEN loss was associated with a non-statistically significant shorter PFS in BRAF-mutant melanoma patients treated with BRAF inhibitor [110]; a recent work affirmed that PTEN-negative melanoma patients have poor outcome as the result of resistance to both TT and ICIs [111].

Activation of the PI3K/AKT/mTOR pathway is an established mechanism of resistance to BRAF and MEK inhibitors; however, it is not so frequent and preclinical models of resistance have shown that it might probably not be enough to drive proliferative escape of melanoma cells [112]. In wild-type *PTEN* melanoma, tyrosine-protein kinase AXL could be responsible for resistance to BRAF and MEK inhibitors by activating AKT [113]. A preclinical study pointed out that AKT inhibition was sufficient to block proliferation of BRAF V600-mutant melanoma cells, even if this effect was stronger when combined with a MEK inhibitor [114], thus suggesting that this approach could be feasible for testing in treatment naïve *BRAF*-mutant patients.

Several clinical trials using specific inhibitors of the PI3K/AKT pathway, alone or in combination with other drugs, are ongoing. PI3K inhibitors currently being tested in melanoma trials are: GSK2636771, in combination with pembrolizumab in PTEN loss metastatic melanoma (Phase 1/2 (NCT03131908) [115]; buparlisib in combination with encorafenib and binimetinib in BRAF-mutant advanced melanoma (NCT02159066) [116]; buparlisib in combination with vemurafenib in BRAF-mutant advanced melanoma (NCT01512251)). However, the latter combination was poorly tolerated, not warranting further study [117]. AKT inhibitor uprosertib has been tested in association with dabrafenib in patients with an advanced tumour, but this combination was not tolerated [118]; another AKT inhibitor, MK2206, is being tested in association with MEK inhibitor selumetinib in advanced melanoma patients who failed therapy with vemurafenib and cobimetinib (NCT01519427). In conclusion, the strong preclinical rationale of combining MAPK and PI3K/AKT inhibitors probably cannot be applied in clinical practice due to excessive toxicities; different approaches and better selection of patients are needed.

### 6.4. Cell Cycle Regulation

Cell cycle is an organized process aimed at cell division through duplication of genetic information, and its activity is aberrant in tumour cells, being a hallmark in human cancer [119]. Among involved molecular machineries, the cyclin-dependent kinase (CDK4/6)-Retinoblastoma (RB) pathway plays the most important role by regulating the step between G0–1 and S phase, determining, in physiological conditions, the duplication of the genome [120]. In response to mitogenic factors, CDK4/6 forms, together with cyclin D, a complex which blocks RB protein through phosphorylation, thus releasing E2F transcription factor; p16INK4a, a protein transcript of *CDKN2A* gene, prevents cyclin D binding by CDK4/6, negatively regulating this phase of cell cycle [121]. Genetic aberrations of members of this pathway are frequent in melanoma, observed in up to 90% of both preclinical and clinical models [122]. Concerning BRAF-mutant melanoma, an over-activated MAPK pathway is responsible for elevated cellular proliferation by enhancing CDK4/6 functions [123]. Metastatic BRAF-mutant melanoma patients with high expression of cell cycle genes (cell cycle signature) compared with those with low expression, showed shorter PFS when treated with BRAF inhibitors alone. However, similar PFS was found with treatment with the combination of BRAF and MEK inhibitors, suggesting a more effective cell cycle blockade by a stronger inhibition of MAPK pathway [124].

Cell cycle components rapidly became an interesting molecular target for anticancer drugs. After the failure of nonselective first-generation CDK inhibitors in solid cancers treatment, due to their dose limiting toxicities as a result of excessive cell cycle inhibition in normal tissues, several CDK4/6 inhibitors were evaluated in solid tumour and melanoma clinical trials, both as single agents or in combination with other drugs [125]. Concerning BRAF-mutant melanoma, preclinical data showed the usefulness of CDK4/6 inhibitor LY2835219 in killing vemurafenib-resistant cells, being the right premise for the use of this class of anticancer drugs in such a clinical scenario [126]. Voruciclib (P1446A-05) has been tested in combination with vemurafenib in a Phase 1 trial (NCT01841463) with good tolerability and efficacy, also in treatment-naïve patients [127]; similar results were reported for ribociclib, a potent and selective CDK4/6 inhibitor, administered together with encorafenib in BRAF-mutant melanoma patients [128]. A preclinical study investigated the optimal timing of adding palbociclib to BRAF and MEK inhibitors, confirming that double inhibition with BRAF and CDK4/6 inhibitors suppresses tumour growth in treatment naïve BRAF-mutant models, whilst the triple inhibition is ineffective after the development of BRAF inhibitor resistance [129]. The LOGIC II trial confirmed that the addition of ribociclib to encorafenib and binimetinib, after progression in advanced BRAF V600-mutant melanoma patients, based on genetic tumour evolution, did not result in a meaningful activity [116]. Therefore, CDK4/6 inhibitors have a central role in the future of BRAF-mutant melanoma, but more efforts in establishing the correct timing and combination with other anticancer agents are needed.

### 6.5. Genomic Instability

Cutaneous melanoma genomes have the highest mutation load of any cancer, mainly attributable to UV radiation signature (C > T nucleotide transitions) [130], resulting in several somatic nonsynonymous mutations which are quantitively measured as number per coding area, a measurement called tumour mutational burden (TMB) [131]. As stated before, a higher TMB has been described in BRAF V600K-mutant cutaneous melanoma with respect to V600E ones, probably because of the higher sun-exposure and the older age of patients harbouring the V600K mutation [22]. Tumour neoantigens, resulting from high TMB, are recognized by neoantigen-specific T cell receptors, eliciting specific anti-tumour immune response and explaining the success of cancer immunotherapy [132].

In a cohort of metastatic melanoma patients treated with a first-line combination of anti-PD-1 and anti-CTLA-4, TMB was notably higher in responder than non-responder patients, and high TMB values were associated with a statistically significant survival benefit [133]; another work analysed response to anti-PD-1 antibodies in melanoma patients with regard to TMB but failed to demonstrate an association between survival and mutational load [134]. BRAF V600-mutant patients treated with first-line anti-PD-1 and anti-CTLA-4 combination therapy reached better overall survival (5-years OS: 60% vs. 48%) than BRAF wild-type ones [69]; however, BRAF-mutant melanoma specimens have shown lower TMB values compared with wild-type ones, suggesting that factors other than neoantigen load are responsible for the good results observed in this specific molecular subtype [135].

Among drugs targeting genome instability in human cancer, poly (ADP-ribose) polymerase (PARP) inhibitors represent the most intriguing ones, alone or in combination with chemotherapy or immunotherapy [136].

PARPs are enzymes that transfer poly(ADP-ribose) from nicotinamide-adeninedinucleotide on target proteins, such as DNA topoisomerases, DNA helicases and base-excision repair factors, in response to DNA single-strand breaks (SSBs) [137]. PARP inhibitors are currently used as maintenance therapy in high-grade serous ovarian cancer patients after platinum-based therapy, with better results obtained in patients harbouring germline or somatic mutations of genes involved in homologous recombination, which is the mechanism involved in double-strand break DNA repair [138]. This molecular subgroup of ovarian cancer patients, known as HR (homologous recombination) deficient, benefit the most from PARP inhibitors since these drugs, blocking the SSB repair, determine the accumulation of DNA breaks, leading cells to apoptosis. Among genetic alterations linked to homologous recombination deficiency (HRD), the most frequent are BRCA1 and 2, RAD51, ATR, ATM, CHK1 and 2, BAP1 and FANCD2 [139].

Concerning melanoma, a study, which explored genetic alterations by NGS in more than 52,000 tumours of 21 different cancer lineages, found HR mutations in slightly less than 20% of melanoma specimens, with BAP1 (7.7%) and ATM (3.7%) being the most frequently mutated genes [140]. In another study, BRAF V600-mutant melanomas harboured fewer HR mutations than BRAF wild-type ones, suggesting a divergence in melanomagenesis between these genetic alterations [141].

PARP inhibitors in combination with chemotherapy have already been studied in unselected melanoma patients, showing a non-clinically significant activity compared to currently approved treatments [142,143].

However, HR mutated melanomas are now under the magnifying glass, with ongoing and future clinical trials using PARP inhibitors in this specific subgroup of patients (niraparib, NCT03925350; olaparib + pembrolizumab, NCT04633902). BRAF-mutant melanoma, independent from HR mutations, could also be a good candidate for PARP inhibitors, as suggested by preclinical studies [144,145].

### 6.6. Epigenetics

Epigenetic mechanisms in the human genome have become a topic of growing interest, especially in cancer research; epigenetic events are alterations in gene expression without a change in DNA sequence: DNA methylation/demethylation, histone modifications, chromatin and nucleosome remodelling are the best-known epigenetic mechanisms involved in carcinogenesis [146]. Concerning melanoma, several epigenetic events have been described in recent years [147]. Focal DNA hypermethylation of tumour suppressor gene promoters, such as PTEN and CDKN2A, have been identified in up to 60% of sporadic melanomas, in some cases also with a prognostic correlation [148,149]; however, many other genes have been found to be methylated in melanoma [150]. The CpG island methylator phenotype (CIMP), a phenomenon originally proposed in colorectal cancer, has also been described in melanoma patients [151]. In preclinical models, the BRAF V600E mutation promotes epigenetic silencing through a transcriptional regulatory pathway which results in hypermethylation [152]; clinical trials investigating potential synergistic effects of ICIs and hypomethylating agents in melanoma patients are currently ongoing [153].

Chromatin-remodelling complexes, of which the switch/sucrose non-fermentable (SWI/SNF) complexes are the most studied, are involved in ATP-dependent compaction and de-compaction of DNA; SWI/SNF complexes are composed of 15 subunits encoded by 29 genes, which are mutated in >20% of human cancers [154]. Concerning melanoma, several studies suggest that SWI/SNF complexes have a key role in genetic regulation during melanocyte differentiation and in melanomagenesis [155]; however, many unanswered questions about the role of these complexes in transcriptional regulation of proliferation and survival genes make this area of research rather fascinating, particularly for new anti-cancer drug design and development [156]. Histone post-translational modifications, mainly acetylation, methylation and phosphorylation, affect chromatin structure. The histone deacetylase (HDAC) family consists of four classes of enzymes, which remove acetyl groups from histones, increasing their positive charge and, as a consequence, enhancing their affinity for DNA [157]. HDAC inhibitors (HDACi), which have been shown to induce histone H3 and H4 acetylation at the CDKN2A gene, thus reactivating expression of p14ARF [148], are under clinical investigation in several kinds of cancer, including melanoma (NCT03765229).

Concerning BRAF-mutant melanoma, many studies have pointed out the role of epigenetic mechanisms, mainly through HDAC, in resistance to BRAF and MEK inhibition [158,159], suggesting also the possibility of preventing the onset of resistance by using HDACi [160]. Indeed, an ongoing trial is investigating the HDACi vorinostat in resistant BRAF V600-mutant advanced melanoma patients (NCT02836548).

EZH2 is a histone methyltransferase which methylates lysine 27 of histone H3 (H3K27), repressing transcriptional activity, and is pathologically activated in approximately 20% of melanoma patients [161]; EZH2 inhibitor tazemetostat is currently being tested alone or in combination with dabrafenib and trametinib in progressive BRAF V600-mutant metastatic melanoma patients (NCT04557956).

In the future, epigenetic regulators will play an important role in this disease, in combination or sequentially with ICIs and/or TT.

### 6.7. Angiogenesis

New vessel formation inside a tumour and its metastasis is a key step in malignant progression, without which the cellular growth would be strongly limited [162]. Glycoproteins of the family of vascular endothelial growth factor (VEGF), especially VEGF-A, which stimulates endothelial cell survival, proliferation and angiogenesis through its binding to VEGF receptor (VEGFR), are upregulated in cancer [163]; the tumour microenvironment—in particular stromal fibroblasts and cancer-associated macrophages—plays a fundamental role in regulating angiogenesis and, as a consequence, in tumour progression [164]. Melanoma has a high angiogenic activity: aggressive melanoma preclinical models show higher levels of VEGF compared to non-aggressive ones [165]. BRAF inhibition could enhance angiogenesis, and consequently melanoma progression, by stimulating cancer-associated macrophages to produce, with a paradoxically activation of the MAPK pathway, VEGF, which stimulates melanoma cell growth [166]. The tumour microenvironment could therefore determine innate resistance to BRAF inhibitors by secreting hepatocyte growth factor (HGF) which activates MAPK and PI3K/AKT through MET receptor [167]. The tyrosine kinase receptor AXL is a member of the TAM family with the high-affinity ligand growth arrest-specific protein 6 (GAS6). The Gas6/AXL signalling pathway is associated with tumour cell growth, metastasis, invasion, epithelial–mesenchymal transition, angiogenesis, drug resistance, immune regulation and stem cell maintenance [168]. As stated before, the Gas6/AXL signalling pathway could activate AKT in wild-type PTEN melanoma [113]; high levels of AXL, together with low levels of microphthalmia-associated transcription factor (MITF), are common in BRAF-mutant melanomas and are associated with early resistance to TTs [169].

Anti-angiogenetic drugs have been tested in melanoma patients but with scarce results so far. Bevacizumab, an anti-VEGF-A antibody, failed to prolong OS as adjuvant treatment in radically resected stage II-III melanoma patients, even if increasing disease-free survival (DFS) [170]; bevacizumab has also been investigated together with dacarbazine in advanced melanoma [171] and together with BRAF and MEK inhibitors in BRAF-mutant metastatic melanoma in a Phase 2 trial (NCT01495988) which was prematurely interrupted. Given the effect of VEGF-targeted agents on the immune system [172], the association of these drugs with immune-checkpoint inhibitors have been explored in several kinds of cancer, including melanoma [173]. To date, bevacizumab is not approved for melanoma treatment, but ongoing clinical trials will elucidate its role in combination with anti-PD-(L)1 drugs (NCT04356729, NCT02681549) [174]. Tyrosine-kinase inhibitors targeting VEGFR, such as lenvatinib and cabozantinib, are currently being tested together with immunotherapy in stage III-IV melanoma (NCT01136967, NCT04091750, NCT03957551) (Table 4).

## 7. Conclusions

BRAF mutational status fills a pivotal role in the management of both advanced and completely resected melanoma patients; thus, special attention should be addressed to the detection of BRAF mutations, with the aim of avoiding the risk and under-treatment of false-negative cases. Deepening knowledge about mechanisms underlying resistance to currently approved therapies that are now being investigated in preclinical and clinical studies, and even going back to molecular pathways, is necessary for improving outcomes in BRAF-mutant patients.

## Figures and Tables

**Figure 1 ijms-22-03474-f001:**
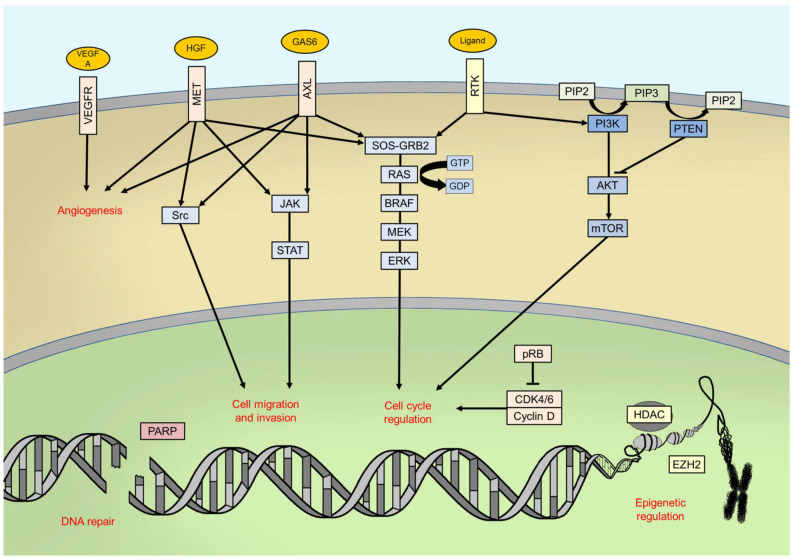
Main intracellular pathways and processes involved in melanomagenesis. RTK: receptor tyrosine kinase; PIP2: phosphatidylinositol 4,5-bisphosphate; PIP3: phosphatidylinositol (3,4,5)-trisphosphate.

**Table 1 ijms-22-03474-t001:** Different clinicopathological characteristics between BRAF V600E- and V600K-mutant melanoma patients.

Mutation	Frequency (among BRAF V600 Mutations)	DFS	PFS (with TT)	TMB	Response to ICIs
V600E	70–88%	Longer	Longer	Lower	Good
V600K	10–20%	Shorter	Shorter	Higher	Better

DFS: disease-free survival; PFS: progression-free survival; TT: targeted therapy; TMB: tumour mutational burden; ICIs: immune checkpoint inhibitors.

**Table 2 ijms-22-03474-t002:** Phase 3 clinical trials in BRAF V600-mutant metastatic melanoma patients.

Phase 3 Clinical Trial	% of BRAF V600-Mutant Patients	Experimental Arm	Standard Arm	Primary Endpoint in BRAF V600-Mutant Patients	Reference
COMBI-d	100%	D + T	D	PFS: 9.3 months vs. 8.8 months (*p* = 0.03)	[59]
COMBI-v	100%	D + T	V	1-year OS: 72% vs. 65% (*p* = 0.005)	[60]
coBRIM	100%	V + C	V	PFS: 9.9 months vs. 6.2 months (*p* < 0.001)	[63,64]
COLUMBUS	100%	E + B	V	PFS: 14.9 months vs. 7.3 months (*p* < 0.0001)	[66,67]
KEYNOTE-006	36.2%	P	I	HR for PFS: 0.44–0.87	[4,69]
CheckMate 067	31%	N + I	I	5-year OS: 60% vs. 30%	[70,71]

D: dabrafenib; T: trametinib; V: vemurafenib; C: cobimetinib; P: pembrolizumab; N: nivolumab; I: ipilimumab; PFS: progression-free survival; OS: overall survival; HR: hazard ratio.

**Table 3 ijms-22-03474-t003:** Adjuvant Phase 3 clinical trials in BRAF V600-mutant high-risk resected melanoma patients.

Phase 3 Clinical Trial	% of BRAF V600-Mutant Patients	Stage (AJCC VII Edition)	Experimental Arm	Standard Arm	Primary Endpoint in BRAF V600-Mutant Patients	Reference
BRIM-8	100%	IIC-IIIB	V	placebo	RFS: 23.1 months vs. 15.4 months (*p* = 0.026)	[74]
COMBI-AD	100%	III	D+T	placebo	4-year RFS: 54% vs. 38%	[73,75]
CheckMate-238	42.1%	IIIB-IV	N	I	HR for 2-year RFS: 0.79	[81]
KEYNOTE-054	43.3%	III	P	placebo	HR for 1-year RFS: 0.59	[80]

V: vemurafenib; D: dabrafenib; T: trametinib; P: pembrolizumab; N: nivolumab; I: ipilimumab; RFS: recurrence-free survival; HR: hazard ratio.

**Table 4 ijms-22-03474-t004:** New therapeutic targets for BRAF V600-mutant melanoma patients and drugs under investigation.

Pathway/Cancer Hallmark	Molecular Target	Drugs under Investigation	Phase of Development	Reference (or NCT Number)
MAPK	ERK1/2	MK-8353	Phase 1 trial	NCT01358331
		Ulixertinib	Phase 1trial	NCT01781429
PI3K-AKT-mTOR	PI3K	GSK2636771	Phase 1/2 trial	[115]
		Buparlisib	Phase 2 trial	[116]
	AKT	Uprosertib	Phase 1 trial	[118]
		MK2206	Phase 2 trial	NCT01519427
Cell cycle regulation	CDK4/6	LY2835219	Preclinical data	[126]
		Voruciclib	Phase 1 trial	[127]
		Ribociclib	Phase 1b/2 trial	[128]
			Phase 2 trial	[116]
		Palbociclib	Preclinical data	[129]
DNA repair	PARP	Rucaparib	Phase 2 trial	[142]
		Veliparib	Preclinical data	[145]
			Phase 2 trial	[143]
		Niraparib	Phase 2 trial	NCT03925350
		Olaparib	Phase 2 trial	NCT04633902
Epigenetic regulation	HDAC	Vorinostat	Phase 1/2 trial	NCT02836548
	EZH2	Tazemetostat	Phase 1/2 trial	NCT04557956
Angiogenesis	VEGF-A	Bevacizumab	Phase 2 trial	NCT04356729
			Phase 2 trial	NCT02681549
	VEGFR, PDGFR	Lenvatinib	Phase 2 trial	NCT01136967
	VEGFR, AXL, MET	Cabozantinib	Phase 2 trial	NCT04091750
			Phase 1b/2 trial	NCT03957551

## Data Availability

The corresponding author will provide the data or will cooperate fully in obtaining and providing the data on which the manuscript is based for examination by the editors or their assignees.

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
