# Peer review of "BRAF Gene and Melanoma: Back to the Future"

_ijms, 2021, doi:10.3390/ijms22073474_

Round 1

Reviewer 1 Report

The review explains a current topic, therefore of interest  to clinicians and all scientific readers.  It is detailed in every part, it is clear and accurate and overall of high quality.

The authors make a systematic contribution to the research literature.

Author Response

We greatly appreciate the First Reviewer's valuable comments and we thank the Reviewer for having grasped the main aim of our manuscript. 

Reviewer 2 Report

Author present here a comprehensive review of the recent literature reporting about the role of BRAF in melanoma. Although I appreciate the focus on past and ongoing clinical trials assessing the role of BRAFi in combination with MEKi and ICi, I would suggest that authors also include/discuss effects of these inhibitors on environmental cells. For example changes in the secretome of tumor cells in response to BRAFi, conseuentially enhancing cell migration. As well, authors should include additional resistance mechanisms e.g. confered by non-genetic mechanisms.

However, these are just suggestions. Minor point that indeed need to be addressed prior to publication:

-RAS proteins are no kinases but GTPases

-authors might include a figure of interactions partners and subsequent signaling processes more just the normal MAPK-axis

-authors should not subsume MET and AXL (ligand of AXL is missing: GAS6) to exclusively control Angiogenesis. This is misleading as both are excellent and well known regulators of cell migration.

-Regarding the role of the different BRAF mutations, authors should include the report about clonal selection in BRAF-driven cancers ("Clonal selection confers distinct evolutionary trajectories in BRAF-driven cancers")

-Authors need to perform a spell check as many typos are still present in the present manuscript

Author Response

We thank the reviewer for her/him time and kind consideration of this manuscript. We have now revised our manuscript to address and accommodate the valuable reviewer’s comments and suggestions. We have included below a point-by-point response to the reviewer’s comments and we hope that the reviewer will find our responses and revisions sufficient.  

  • I would suggest that authors also include/discuss effects of these inhibitors on environmental cells. For example changes in the secretome of tumor cells in response to BRAFi, consequentially enhancing cell migration. As well, authors should include additional resistance mechanisms e.g. confered by non-genetic mechanisms.
    • We greatly appreciate the valuable reviewer's suggestions, however we have tried to focus our review on the molecular pathways that are now being investigated in at least Phase I clinical trials. Since the data reported are so many, we would like to leave the structure and content of our manuscript unchanged, however we will seriously consider the reviewer's suggestions in the development of our future papers. 
  • -RAS proteins are no kinases but GTPases
    • We thank the reviewer for the careful observation and we have modified the text, accordingly (Page 2, line 57). Please, see the tracked-changes of our manuscript. 

  • -authors might include a figure of interactions partners and subsequent signaling processes more just the normal MAPK-axis
    • We appreciate the reviewer's suggestion and we have updated the previous figure into a more complete version. Please, see the tracked-changes of our manuscript (Figure 1). 
  • -authors should not subsume MET and AXL (ligand of AXL is missing: GAS6) to exclusively control Angiogenesis. This is misleading as both are excellent and well known regulators of cell migration.
    • We apologize for the superficial description of the MET and AXL pathways. We completely share the reviewer's comment and we have added more details in the Angiogenesis paragraph (paragraph 6.7 from line 883), updating contextually also the bibliography. Please, see the tracked-changes of our manuscript. 
  • -Regarding the role of the different BRAF mutations, authors should include the report about clonal selection in BRAF-driven cancers ("Clonal selection confers distinct evolutionary trajectories in BRAF-driven cancers").
    • We greatly appreciate this valuable suggestion and thank the reviewer for having highlighted this important paper, that we have cited in our manuscript in the paragraph n. 3: BRAF mutations in melanoma: epidemiology and clinic-pathological correlations. Please, see the tracked-changes of our manuscript. 
  • -Authors need to perform a spell check as many typos are still present in the present manuscript.
    • We thank the reviewer for this observation and we have performed, as suggested, a spell check. We hope to have corrected all the previous typos. Please, see the tracked-changes of our manuscript. 

Reviewer 3 Report

The manuscript regarding BRAF mutations in melanoma review is very well presented with impressive amount of clinical and molecular data and information. There are only few minor things for authors to be addressed:

Ras proteins should be referred as GTPases, not kinases (page 2, line 57).

The VE1 monoclonal antibody has many faults that should be discussed in more details (e.g. many false positives and low specificity in primary melanomas; not suitable for acral lentiginous melanoma etc).

The reference 'Marchant et al., 2014' (page 5, line 194) is not listed in the reference list, and this reference in the text should be formatted as a number.

Author Response

We would like to express our great appreciation to you for your careful and precious observations on our manuscript. We have studied your comments carefully and have made a revision which was marked in the revised paper. The content list below is the brief summarization point to point. 

  • Ras proteins should be referred as GTPases, not kinases (page 2, line 57).
    • Of course, We share the reviewer’s comment and we have modified the text accordingly. Please, see the tracked-changes version of our manuscript.
  • The VE1 monoclonal antibody has many faults that should be discussed in more details (e.g. many false positives and low specificity in primary melanomas; not suitable for acral lentiginous melanoma etc).
    • We thank the reviewer for this comment and we have added in the manuscript more details about the VE1 monoclonal antibody, in the Immunohistochemistry paragraph. Please, see the tracked-changes version of our manuscript.
  • The reference 'Marchant et al., 2014' (page 5, line 194) is not listed in the reference list, and this reference in the text should be formatted as a number.
    • We greatly appreciate this careful observation and the reference has been added in the reference list and in the text with number 54. Please, see the tracked-changes of our manuscript.